# Application of RNA-Seq Technology for Screening Reproduction-Related Differentially Expressed Genes in Tibetan and Yorkshire Pig Ovarian Tissue

**DOI:** 10.3390/vetsci11070283

**Published:** 2024-06-21

**Authors:** Yikai Yin, Jian Zhang, Xindi Li, Mengqi Duan, Mingxuan Zhao, Feifan Zhang, Yangzom Chamba, Peng Shang

**Affiliations:** 1College of Animal Science, Tibet Agriculture and Animal Husbandry College, Linzhi 860000, China; yyk0426@126.com (Y.Y.); 13989046293@126.com (J.Z.); lixindi75@136.com (X.L.); zduanduan0117@163.com (M.D.); m19863998873@163.com (M.Z.); 17654540263@163.com (F.Z.); 2The Provincial and Ministerial Co-Founded Collaborative Innovation Center for R&D in Tibet Characteristic Agricultural and Animal Husbandry Resources, Linzhi 860000, China; 3Key Laboratory for the Genetic Improvement and Reproduction Technology of the Tibetan Swine, Linzhi 860000, China

**Keywords:** RNA-Seq, Tibetan pig, reproductive traits, ovarian tissue

## Abstract

**Simple Summary:**

The problem of low fertility in Tibetan pigs has a long history. In order to solve the problem of low fertility in Tibetan pigs, in this study, we successfully constructed cDNA libraries of the transcriptome from ovarian tissues of Tibetan and Yorkshire pigs using RNA-Seq technology, analyzed and screened the genes significantly enriched in the pathways closely related to the regulation of reproduction, and finally identified *AR*, *CYP11A1*, *CYP17A1*, *INHBA*, *ARRB2*, *EGFR*, *ETS1*, *HSD17B1*, *IGF1R*, *MIF*, *SCARB1*, and *SMAD4* as important candidate genes. The 12 differentially expressed genes screened in relation to reproduction were validated using RT-qPCR. The results showed that six genes were highly significantly (*p* < 0.01) higher in expression in Tibetan pigs than in Yorkshire pigs, i.e., *AR*, *CYP17A1*, *EGFR*, *ETS1*, *IGF1R*, and *SMAD4;* on the other hand, six genes were highly significantly (*p* < 0.01) lower in expression in Tibetan pigs than in Yorkshire pigs, i.e., *CYP11A1*, *INHBA*, *ARRB2*, *HSD17B*, *MIF*, *SCARB1*. This study provides a theoretical basis for improving the regulatory genes in Tibetan pigs.

**Abstract:**

The purpose of this study was to explore and verify genes that regulate the reproductive traits of Tibetan pigs at the mRNA level. The ovarian tissues of Tibetan pigs (TPs) and Yorkshire pigs (YPs) were selected as research objects, and cDNA libraries of the ovarian tissue transcripts of Tibetan pigs and Yorkshire pigs were successfully constructed by the RNA-Seq technique. A total of 651 differentially expressed genes (DEGs) were screened, including 414 up-regulated genes and 237 down-regulated genes. Through GO and KEGG enrichment analysis, it was found that these differentially expressed genes were significantly enriched in cell process, reproductive process, reproduction, cell proliferation, binding, and catalytic activity, as well as oxidative phosphorylation, endocrine resistance, thyroid hormone, Notch, and other signal transduction pathways. Genes significantly enriched in pathways closely related to reproductive regulation were analyzed and selected, and the *AR*, *CYP11A1*, *CYP17A1*, *INHBA*, *ARRB2*, *EGFR*, *ETS1*, *HSD17B1*, *IGF1R*, *MIF*, *SCARB1*, and *SMAD4* genes were identified as important candidate genes. Twelve differentially expressed genes related to reproduction were verified by RT-qPCR. The results showed that the expression of the *AR*, *CYP17A1*, *EGFR*, *ETS1*, *IGF1R*, and *SMAD4* genes was significantly higher in Tibetan pigs than in Yorkshire pigs, while the expression of the *CYP11A1*, *INHBA*, *ARRB2*, *HSD17B*, *MIF,* and *SCARB1* genes in Tibetan pigs was significantly lower than in Yorkshire pigs. The purpose of this study is to provide a theoretical basis for exploring the molecular mechanism of reproductive trait effect genes and the application of molecular breeding in Tibetan pigs.

## 1. Introduction

The Tibetan pig (TP) is a unique local pig breed in the high-altitude area of the Qinghai–Tibet Plateau of China. The Tibetan pig is robust, with a long pointed mouth, fine hooves, hard bones, black and smooth fur, good running ability, especially developed cardiopulmonary function, and strong resistance to disease and cold. Tibetan pork has thin skin, tender meat scent, fine muscle fibers, density, and uniform muscle fat distribution, and it is rich in nutrients. It is especially rich in protein and amino acids, with protein and amino acid contents of up to 81% and 73% per 100 g of dry lean meat, and the linoleic acid content is 2.5 times higher than the average domestic pig.

The tender and nutritious meat of Tibetan pigs is very popular among people, and the demand for Tibetan pork is increasing day by day. However, because the main feeding methods of Tibetan pigs are grazing and semi-grazing, their feeding cycle is long, and the high-altitude environment of the Tibetan Plateau leads to their low reproductive performance. Therefore, improving the reproductive performance of Tibetan pigs is of positive significance for local economic development and the conservation of high-quality genetic resources [1]. Most fecundity traits belong to quantitative traits, which are regulated by some major genes and many minor genes [2]. These genes that regulate reproductive traits are called fertilization genes. Mammalian ovarian development is affected by many factors, including species differences, external environment, hormone levels, gene expression levels, regulation of related pathway regulation, etc. [3]. These factors affect whether ovarian function is normal or not, and often determine the level of reproductive performance [4]. In the reproductive system, the ovary has a series of functions, such as periodic synthesis and secretion of steroids, proliferation of granulosa cells and membrane cells [5], follicular development, production and excretion of oocytes, and formation of corpus luteum [6]. It is an important organ in maintaining reproductive activity and regulating the reproductive process [7].

The *BMP15* and *INHBA* genes have been reported to have some level of mRNA expression during the estrous cycle in sheep versus in the sinus follicle GC. The role of *INHBA* in follicular maturation and steroidogenesis in the little-tailed ewe is associated with an increase in litter size [8], while *BMP15* is expressed in oocytes during the final development stages in sheep [9] and regulates GC proliferation and differentiation, with *INHBA* and *BMP15* gene expression up-regulated.

Recombinant resistance (0.1, 1, 10, and 100 ng/mL) to steroid hormones (progesterone, androstenedione, testosterone, and estradiol) in sows awaiting sexual maturity stimulated steroidogenesis within the follicle, and the expression of the cytochrome family genes [10], *CYP11A1* and *CYP17A1*, was increased. *INHBA* levels were positively correlated with the secretion of activin and inhibin A in culture medium. However, both knockdown and overexpression of *INHBA* decreased the expression of the inhibin subunit alpha (INHA) [11]. High-throughput RNA sequencing (RNA-Seq) technology has become a popular tool for transcriptome analysis and exploration of unknown genes [12]. RNA-Seq technology is increasingly being used in studies related to porcine reproductive performance to help identify transcriptomic datasets and characterize functional candidate genes. Many studies using RNA-Seq technology have identified several DEGs associated with porcine fertility [13]. In this study, in order to select DEGs associated with Tibetan and Yorkshire pig reproduction for a better understanding of the molecular factors involved in the porcine estrous cycle and their regulatory genes, the expression profiles of mRNA in the ovaries of adult Tibetan pigs and adult Yorkshire pigs were compared using RNA-Seq technology [14]. Their differentially expressed genes were identified. They were also analyzed by GO enrichment and the KEGG pathway, in order to obtain the set of significantly differentially expressed genes and select important candidate genes from them [15]. This will provide some theoretical basis for future exploration of molecular regulatory mechanisms related to reproductive traits and molecular markers to improve the reproductive performance of Tibetan pigs [16]. These results will provide new information on the molecular mechanisms of the estrous cycle in sows and provide data and references for the study of reproductive genes in sows [15].

## 2. Materials and Methods

### 2.1. Laboratory Animal Ethics Statement and Ovarian Tissue Sampling

Tibetan pigs and Yorkshire pigs were selected as the animals in this experiment. These animals were not endangered or protected. They were sacrificed humanely to minimize suffering. A total of 12 sows from the two breeds, TPs and YPs, were reared at the same level of husbandry in the pasture of the Tibetan College of Agriculture and Livestock. After 180 days of rearing, we collected their ovarian tissues and froze them rapidly in liquid nitrogen. These tissues were used for total RNA extraction. All experimental research was carried out in strict accordance with the guidelines approved by the Animal Welfare Committee of the Tibetan College of Agriculture and Animal Husbandry.

### 2.2. The Process of RNA Extraction

Total RNA was extracted from ovarian tissues using TRIzol reagent (Invitrogen, San Diego, CA, USA). The integrity and total amount of the RNA samples were then accurately determined using an Agilent 2100 bioanalyzer (Agilent Technologies, Santa Clara City, CA, USA). Sequencing libraries were generated using the NEBNext^®^ UltraTM RNA Library Prep Kit for Illumina^®®^ (NEB, Ipswich, MA, USA) according to the manufacturer’s instructions. After library construction, initial quantification was performed using a Qubit 2.0 Fluorometer to dilute the library to 1.5 ng/μL. Subsequently, the size of the libraries was tested using an Agilent 2100 bioanalyzer. After the insert size was as expected, the effective concentration of the libraries was determined by RT-qPCR. The effective concentration of the library was accurately quantified by RT-qPCR (the effective concentration of the library was greater than 2 nM) to ensure the quality of the library.

### 2.3. Analysis of Sequencing Data

The raw data obtained from sequencing contain a small number of reads with sequencing adapters or low-quality sequences, which need to be filtered in order to ensure the quality and reliability of data analysis. Filtering stages: remove reads with adapters; remove reads containing N (N means that the information of the bases could not be determined); and remove low-quality reads (Qphred < 20 bases accounts for more than 50% of the total length of the reads). After raw data filtering, sequencing error rate verification, and GC content distribution verification, clean reads were obtained for subsequent analysis. All downstream analyses were based on clean high-quality data. The mapping of the reference genome was performed using HISAT2 v2.1.0 (Sus scrofa genome v11.1, downloaded from ENSEMBL Web server). The reads assigned to each gene were calculated using the feature Counts (1.5.0-p3). The FPKM for each gene was then calculated based on the length of the gene, and the number of reads assigned to that gene was calculated. FPKM refers to the expected number of fragments per kilobase per million base pairs of sequenced transcript fragments. DEGs between TP group ovarian tissue and YP group ovarian tissue were identified using the following filtering criteria: padj < 0.05, |log_2_FoldChange| > 0.5.

### 2.4. Gene Ontology and Pathway Enrichment Analysis

The GO and KEGG pathway enrichment analysis of differentially expressed genes was performed by Omicshre Tools (www.omicshare.com, accessed on 13 November 2023). GO is a comprehensive database describing gene functions, which can be categorized into three types: biological process (BP), cellular component (CC), and molecular function (MF). GO functional enrichment uses padj < 0.05 as the threshold for significant enrichment. KEGG is a database resource for understanding biological systems from molecular-level information, especially large-scale molecular datasets generated by genome sequencing and other high-throughput databases. High-throughput databases are used to understand the high-level functions and utility of biological systems such as cells, organisms, ecosystems, etc. KEGG pathways with padj < 0.05 were considered significantly enriched.

### 2.5. Real-Time Fluorescence Quantitative PCR (RT-qPCR) Validation

To validate the results of the sequencing analysis, we selected 12 mRNAs for the RT-qPCR analysis (Appendix A). Total RNA from each ovary sample was extracted using TRIzol reagent (Invitrogen, Carlsbad, CA, USA). The chamQ SYBR Color qPCR Master Mix (2×) with a 2 µL cDNA template and upstream and downstream primers was used. The reaction program was as follows: 95 °C for 5 min (1 cycle), 95 °C for 30 s, 56 °C for 30 s, and 72 °C for 40 s (40 cycles). The sequences of the internal primers for RT-qPCR are shown in Table 1.

## 3. Results

### 3.1. Total RNA Quality Assay

The extraction of RNA from the test samples is shown in Figure 1; the baseline RNA of each sample is flat, without genomic contamination, the concentration of RNA is more than 100 ng/μL, the total amount of RNA is more than 3 μg, and the integrity value is more than 7.0, indicating that the RNA extracted from the test samples meets the above standards of transcriptome sequencing after examination.

### 3.2. Status of Transcriptome Sequencing Data

The statistics of the bases of the downlinked data in this experiment are shown in Table 2, which shows that the total number of fragments with data was in the range of 42.2–48.9 million, and the mapped fragment comparison rate of each sample was above 94%. The distribution of A, T, C, and G bases in each sample was uniform and there was no separation of AT and GC. The distribution of exons and introns in the cDNA libraries of the TP and YP groups was successfully localized to the sus scrofa11.1 reference genome, and the proportion of exon regions was more than 70%. In conclusion, the quality of the sequencing data is good and meets the requirements for subsequent experiments and analysis.

### 3.3. Differentially Expressed Gene (DEG) Analysis

A total of 70,753 genes were detected in the 12 libraries, of which two groups expressed a total of 14,404 genes (see Figure 2a), and gene expression levels were assessed using the FPKM method. The FPKM values of the 12 library samples were clustered using a hierarchical clustering method, and their rows were averaged and scored (Z score). Heat maps of gene correlations between samples were analyzed and plotted (see Figure 2b), which showed that the general differences in the expression of all genes were not significant and the correlation of gene expression between samples was good, reflecting the reliability of the experiments and sample selection, which can be applied to subsequent differential gene screening and analysis. To analyze the differences in the transcriptome between the TP ovarian tissues and the YP ovarian tissues, the YP and TP groups were compared. A total of 651 significant DEGs were identified under the screening condition of |log_2_FoldChange| > 0.5, *p* < 0.05, of which 237 genes were down-regulated and 414 genes were up-regulated (see Figure 2c,d). Details of the first 20 genes up-regulated and down-regulated are given in Appendix A.

### 3.4. GO Functional Enrichment Analysis of Differentially Expressed Genes

In order to further extend the molecular characterization of DEGs, DEGs were annotated using GO terms from the GO database. The GO terms were divided into three categories, including biological process, molecular function, and cellular component. As illustrated in Figure 3, the initial five annotations for each category are as follows: (1) biological process: cellular process, biological regulation, metabolic process, regulation of biological process, and organization of the cellular component or biogenesis; (2) molecular function: binding, catalytic activity, molecular function regulator, transcription regulator activity, and molecular transporter activity; (3) cellular components: cell, cell part, organelle, organelle part, and membrane.

### 3.5. Enrichment Analysis of the KEGG Pathway for Differentially Expressed Genes

By integrating transcriptomic data with bioinformatics analysis, we identified the following differentially expressed genes as potential candidates for further validation. *AR* genes are significantly enriched in the oocyte meiotic pathway, and their mediated signaling plays a physiologically important role in the female reproductive system, regulating follicle growth and expression. The *ARRB2* and *ETS1* genes are involved in endocytosis, cell senescence, the MAPK signaling pathway, and the signal transduction pathway. The *SCARB1*, *CYP11A1*, and *CYP17A1* genes are enriched in metabolism, aldosterone synthesis and secretion, and the cortisol synthesis and secretion signaling pathway. They are synergistic in regulating steroid hormone synthesis and secretion, and promoting normal follicle growth. The *SCARB1* gene is expressed primarily in steroidogenic tissues and functions as a key transporter gene on the cell membrane of precursor cells involved in steroid synthesis. *EGFR* and *IGF1R* are growth factor receptors that are significantly enriched in the autophagy and PI3K-Akt signaling pathways. The *EGFR* gene activates the EGFR-PI3k-Akt signaling pathway and interferes with key components of the signaling pathway, thus inhibiting cell growth. The IGF1R gene is involved in the mediation of the IGF gene, which exerts a mitogenic effect and participates in mitosis during the cleavage process, thereby increasing protein synthesis. The *HSD17B1* and *MIF* genes are enriched in the metabolic pathway and the ovine steroidogenic pathway, respectively. *HSD17B1* acts as a catalytic enzyme to catalyze the final step of estradiol biosynthesis and is an important gene in the synthesis and metabolism of steroid hormones. The expression pattern of the *MIF* gene in porcine follicles of different diameters was studied, with the hypothesis that this gene may be related to follicular development and ovulation. The *INHBA* gene is enriched in the Hippo signaling pathway, exerting a promotive effect on follicle-stimulating hormone. Figure 4 shows the *SMAD4* gene is enriched in the adherence junction and cell cycle pathways, which is an essential regulator gene for cell differentiation, migration, and apoptosis. The KEGG database is mainly enriched in the axon guidance, Alzheimer’s disease, oxidative phosphorylation, apoptosis, Parkinson’s disease, hepatocellular carcinoma, and non-alcoholic fatty liver disease pathways.

### 3.6. RT-qPCR Validation Results

By combining functional annotation with results from the signaling pathways and previous studies to carry out further analysis and screening of DEGs, 12 genes, *AR*, *CYP11A1*, *CYP17A1*, *INHBA*, *ARRB2*, *EGFR*, *ETS1*, *HSD17B1*, *IGF1R*, *MIF*, *SCARB1*, and *SMAD4*, were selected among the results of the RNA sequencing to validate the results (Table 3), including 6 genes up-regulated and 6 genes down-regulated. Figure 5 shows the RT-qPCR results of the 12 selected genes; it can be seen that the expression trend of the 12 genes is consistent with the RNA-seq results.

## 4. Discussion

In this study, we analyzed transcriptome data from Tibetan and Yorkshire pigs and obtained a total of 14,404 genes that were expressed in both breeds of pigs. Functional annotation of significantly differentially expressed genes revealed that a large number of genes were involved in cellular processes, reproductive processes, cell proliferation, catalytic activity, and other functional classifications closely related to the reproduction process. The results of KEGG enrichment showed that there were multiple signaling pathways, and that different signaling pathways crossed each other, forming a complex signaling network system in the ovary. Genes with different degrees of up- and down-regulation were commonly enriched in reproduction and reproduction-related signaling pathways, including oxidative phosphorylation, endocrine resistance, thyroid hormone, tight junction, autophagy, Notch, and other important pathways related to reproductive traits. Existing studies have shown that the above signaling pathways can regulate ovulation and embryonic development in females, or produce a series of biological effects, such as promoting cell growth and proliferation and mediating apoptosis, to regulate mammalian reproductive activities [8,17,18,19], which are important for the reproductive physiology of organisms. The results not only confirm the precision of this study but also help us to screen reproduction-related genes and analyze the up- and down-regulation of their expression.

*AR* genes are significantly enriched in the oocyte meiotic pathway, and their mediated signaling plays an important physiological role in the female reproductive system [20,21,22], regulating follicle growth and expression. *ARRB2* and *ETS1* genes are involved in endocytosis, cell senescence, and signaling pathways of the MAPK signaling pathway. The *CYP11A1* gene and the *CYP17A1* gene co-regulate, and they play a key role in the synthesis and secretion of steroid hormones and in the promotion of follicle regeneration, etc. [23]. The *SCARB1* gene, mainly expressed in steroidogenesis tissues, is a key transporter gene for the cell membrane of the precursor of steroid synthesis. EGFR and *IGF1R* are growth factor receptors that are significantly enriched in the autophagy and PI3K-Akt signaling pathways, intervene in key components of the signaling pathway, and, thus, inhibit cell growth [24]. The *IGF1R* gene, which plays a mitogenic role, participates in mitosis during oogenesis and increases protein synthesis [8,16]. The *HSD17B1* gene and *MIF* gene are enriched in both the metabolic pathway and the ovarian steroidogenic pathway. Chen et al. studied the expression pattern of the *MIF* gene in porcine follicles of different diameters, and hypothesized that this gene could be related to follicular development and ovulation. The *INHBA* gene was enriched in the Hippo signaling pathway, and has a promotional effect on follicle-promoting hormone. The *SMAD4* gene was enriched in the adherence junction and cell cycle pathways, and is an essential regulator of cell differentiation, migration, and apoptosis.

The results of the RT-qPCR validation test demonstrated that the expression of 12 genes, *AR*, *CYP11A1*, *CYP17A1*, *INHBA*, *ARRB2*, *EGFR*, *ETS1*, *HSD17B1*, *IGF1R*, *MIF*, *SCARB1*, and *SMAD4* [25], in ovarian tissues of Tibetan and Yorkshire pigs exhibited highly significant differences and was correlated with ovarian function. The mRNA expression levels of the genes *CYP11A1*, *INHBA*, *ARRB2*, *HSD17B*, *MIF*, and *SCARB1* in the ovarian tissues of the Yorkshire pig were all found to be significantly higher than those of the Tibetan pig. The synthesis of steroid hormones in the mammalian ovary initially utilizes cholesterol as a substrate, with progesterone, androgen, and estrogen being synthesized in a sequential manner by the synergistic action of follicular membrane cells and granulosa cells. The *CYP11A1* and *CYP17A1* genes play a regulatory role in the synthesis of these hormones, with the *CYP11A1* gene acting as a catalytic enzyme in the synthesis of pregnenolone, the initial step in the hormone synthesis reaction. The enzyme of side-chain cleavage encoded by the *CYP11A1* gene is the key catalytic enzyme in pregnenolone production, the initial step of the hormone synthesis reaction, the first step of the hormone synthesis reaction, and the first step of the hormone synthesis reaction [26]. The *CYP11A1* gene encodes the cholesterol side-chain cleavage enzyme, which is the key catalytic enzyme for the generation of pregnenolone, the initial step of hormone synthesis, and the conversion of progesterone to androstenedione by cytochrome P450 17α-hydroxylase encoded by the *CYP17A1* gene, which is the intermediate step of steroid hormone synthesis. It is evident that the *CYP11A1* and *CYP17A1* genes play a pivotal role in the normal development of ovarian tissue [27]. Wang et al. demonstrated that the *CYP11A1* and *CYP17A1* genes exhibited synchronization in response to varying concentrations of follicle-stimulating hormone (FSH), influencing the expression of genes synthesized by steroid hormones in bovine luminal follicular granulosa cells. The results of this experiment demonstrated that the level of *CYP17A1* gene mRNA was significantly higher in the ovarian tissues of Yorkshire pigs than in those of Tibetan pigs. This also corroborates the hypothesis that these two genes act in a synergistic manner to regulate ovarian tissue development and play a pivotal role in the normal functioning of the ovary. The *ARRB2* gene belongs to the Arrestin family of regulatory proteins, which affects cell proliferation and apoptosis by regulating the signaling pathway mediated by mitogen-activated protein kinase (MAPK) [28,29]. Macrophage migration inhibitory factor (*MIF*) is secreted by various immune cells and is involved in cellular immunity by inhibiting macrophage migration to the site of inflammation. Matsu F et al. found that during follicular development, many follicles mature or become atretic and degenerate, which triggers a localized inflammatory response in the ovary [25]. Yang F et al. found that the expression of the *MIF* gene was significantly higher in healthy follicles than in atretic follicles of the same size in sows. In healthy and atretic follicles of the same size in sows, the expression of the *MIF* gene in healthy follicles was found to be significantly higher than that in atretic follicles [30]. This suggests that *MIF* genes play a role in follicular development and regulate the immune response. From the above, it can be seen that both the *ARRB2* and *MIF* genes are involved in the regulation of the cell cycle, as well as follicular growth and development, and influence the function of the ovary.

*HSD17B* is a key gene for total steroid hormone synthesis in ovarian tissue along with the *SCARB1* gene. Progesterone and estrogen are steroid hormones that play an important role in the regulation of follicular development. The *HSD17B* gene is the most common isoform of the short-chain dehydroreductase family and acts as a catalytic enzyme in the synthesis of steroid hormones [31,32]. Hakkarainen et al. found that knockdown of the *HSD17B* gene in mice resulted in abnormal granulosa cell morphology, impaired follicular luteinization, and decreased reproductive capacity [33]. The *SCARB1* gene is a high-effector gene, the first functionally active gene capable of promoting selective uptake of HDL-C, predominantly expressed in steroidogenic tissues, and a key transporter gene of precursors of steroid synthesis in the cell membrane [34]. In the present study, the *SCARB1* gene was not only highly significantly different between the two breeds of pigs, but was also highly expressed in ovarian tissues. It can be hypothesized that both the *HSD17B* and *SCARB1* genes play an important role in the regulation of the process of steroid synthesis in ovarian tissues, and the relationship between the two needs to be further verified.

The results of this study indicate that the expression of the *AR*, *CYP17A1*, *EGFR*, *IGF1R*, *ETS1*, and *SMAD4* genes in the ovarian tissues of Tibetan pigs is significantly higher than that of Yorkshire pigs. Furthermore, the expression of *EGFR* and *IGF1R*, which are the same growth factor receptors, is significantly enriched in the autophagy and PI3K-Akt signaling pathways. The *EGFR* gene can activate the EGFR-PI3K-Akt-related signaling pathway, thereby interfering with key components of the pathway and inhibiting cell growth [35]. The *IGF1R* gene can mediate the *IGF* gene, which exerts a mitogenic effect, participating in the mitosis of the egg cleavage process and increasing protein synthesis. During the mammalian estrous cycle, ovarian tissue undergoes a series of morphological and functional changes, including oocyte maturation and egg expulsion, as well as the formation, maintenance, and degeneration of the corpus luteum [36]. These processes entail the dynamic equilibrium between cell proliferation, apoptosis, reorganization, and degradation, among other factors. The *ETS1* gene is a member of the Ets family of transcription factors and is expressed in a variety of tissues, including vascular endothelial cells, mammary epithelial cells [37], and other tissues. Previous studies have demonstrated that the *ETS1* gene is involved in the regulation of a multitude of biological processes, including cell proliferation and differentiation, apoptosis, angiogenesis, and embryonic development. *SMAD4* is a member of the TGF-β/Smad family of transcription factors. *SMAD4* is the sole co-mediated gene in both the TGF-β/Smad and BMP/Smad signaling pathways. It plays a pivotal role in the biology of mammalian ovarian granulosa cells [38]. Wang W et al. demonstrated that *SMAD4* knockdown in porcine follicular granulosa cells resulted in the blockage of granulosa cell proliferation and an increase in apoptosis [39]. Specifically, *SMAD4* knockdown in mouse ovaries led to severe oocyte abnormalities and the premature luteinization of granulosa cells, as well as infertility [40]. The above studies demonstrate that the *SMAD4* gene plays a pivotal role in the normal development of the ovary. Previous studies have also demonstrated that the functions of the *EGFR*, *IGF1R*, *ETS1,* and *SMAD4* genes are closely related to the development of the ovary.

## 5. Conclusions

In this study, we successfully constructed transcriptome cDNA libraries from Tibetan and Yorkshire pig ovarian tissues using RNA-Seq technology and obtained a total of 651 significantly differentially expressed genes in Tibetan and Yorkshire pigs by |log_2_FoldChange| > 0.5, padj < 0.05. Of these, 414 were up-regulated and 237 were down-regulated. The expression of the *AR*, *CYP17A1*, *EGFR*, *ETS1*, *IGF1R*, and *SMAD4* genes in the ovarian tissues of Tibetan pigs was found to be significantly higher than that of Yorkshire pigs (*p* < 0.01). Conversely, the expression of the *CYP11A1*, *INHBA*, *ARRB2*, *HSD17B*, *MIF*, and *SCARB1* genes in the ovarian tissues of Tibetan pigs was significantly lower than that of Yorkshire pigs (*p* < 0.01). The mRNA expression levels of the 12 candidate genes were verified by RT-qPCR, which revealed significant or highly significant differences in the expression of these genes in the ovarian tissues of Tibetan and Yorkshire pigs. Furthermore, the functions of the genes were found to be closely related to the regulation of reproduction traits, and the gene expression trends were consistent with the transcriptomic data, which proved the validity and authenticity of the transcription sequencing. This study provides a theoretical basis for improving the reproductive performance of Tibetan pigs.

## Figures and Tables

**Figure 1 vetsci-11-00283-f001:**
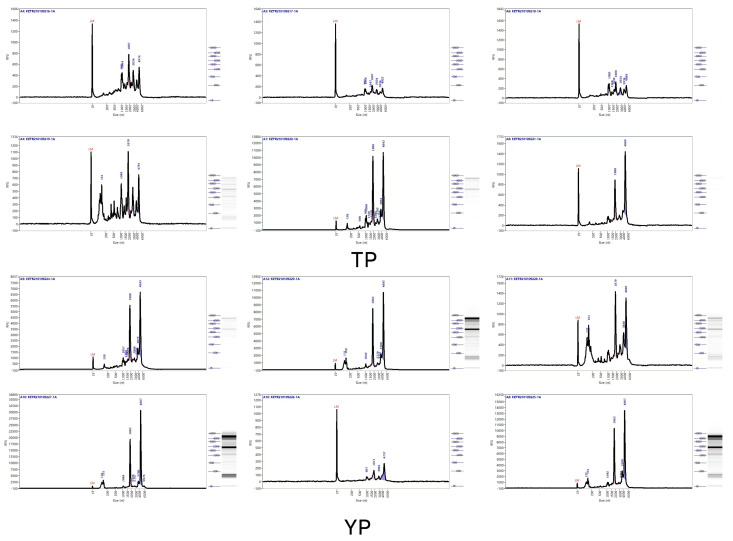
RNA detection from sequenced samples. Note: 12 samples are shown in the QC peak chart; 16 samples were sent, including 8 Tibetan pigs and 8 Yorkshire pigs, of which 4 cases were backup samples of 2 Tibetan pigs and 2 Yorkshire pigs; 12 cases were tested and analyzed; and 12 samples of higher quality were selected. The first 6 are Tibetan pig samples and the last 6 are Yorkshire pig samples.

**Figure 2 vetsci-11-00283-f002:**
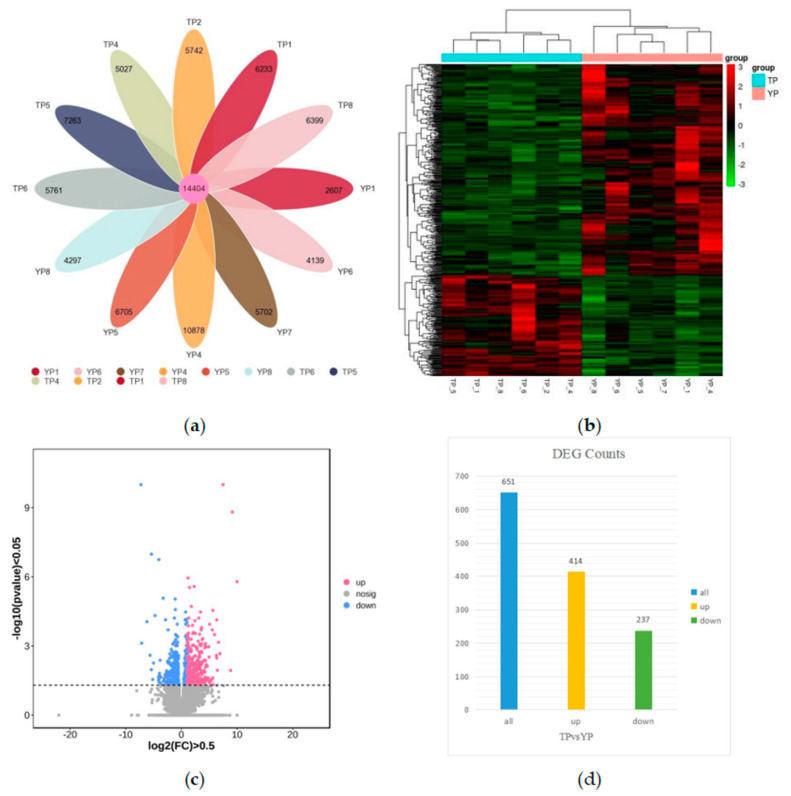
Results of differential gene screening. (**a**) TP vs. YP coexpressed gene mapping, (**b**) differentially expressed gene clustering heat map, (**c**) volcano mapping of differentially expressed genes and (**d**) histogram of the number of differentially expressed genes.

**Figure 3 vetsci-11-00283-f003:**
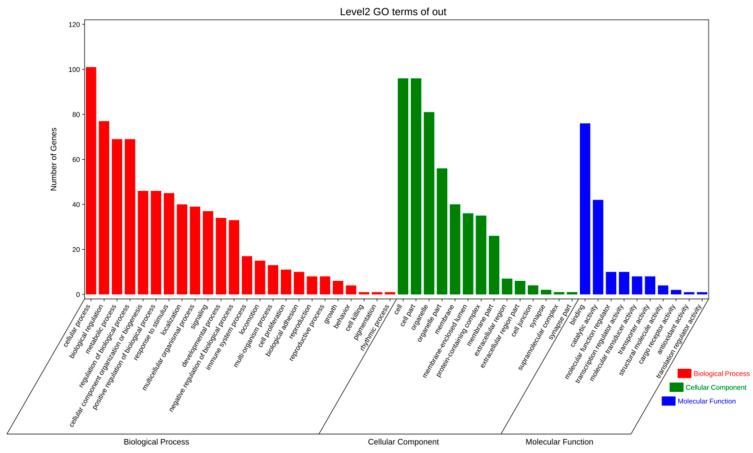
Significantly enriched entry for differentially expressed TOP50 genes.

**Figure 4 vetsci-11-00283-f004:**
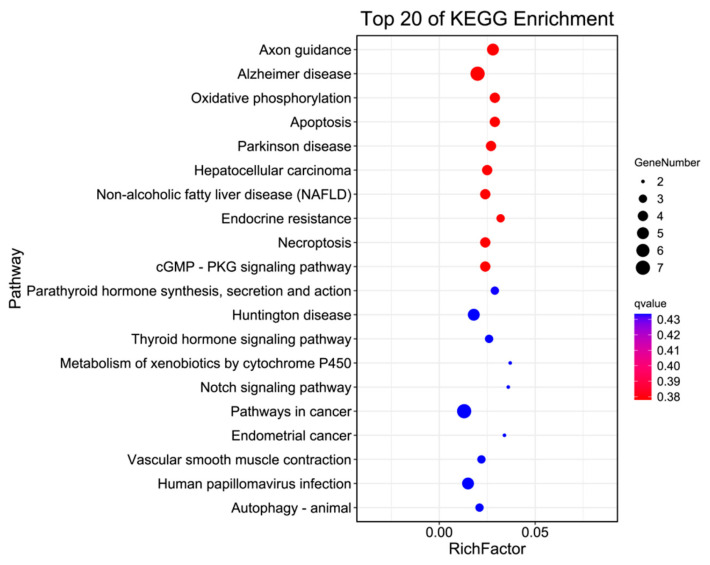
TOP 20 pathways with significant KEGG enrichment.

**Figure 5 vetsci-11-00283-f005:**
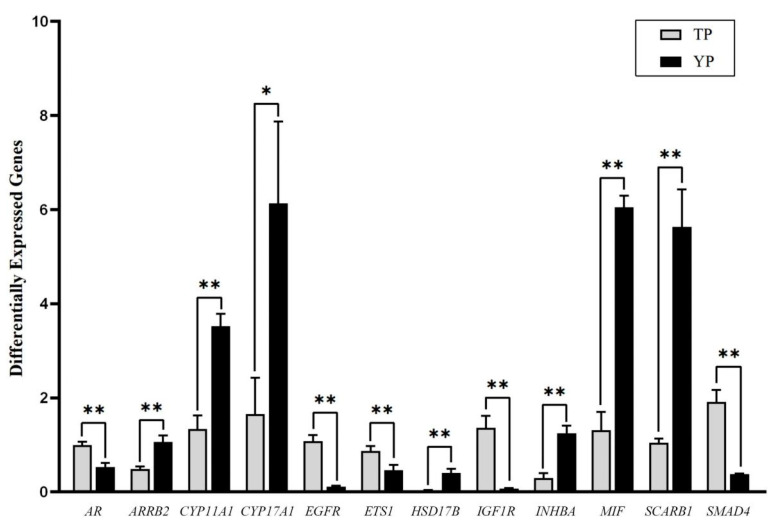
Validation of RNA-seq of *AR*, *CYP17A1*, *EGFR*, *ETS1*, *IGF1R*, *SMAD4, CYP11A1*, *INHBA*, *ARRB2*, *HSD17B*, *MIF*, and *SCARB1* genes by quantitative real-time polymerase chain reaction. Note: * is a significant difference (*p* < 0.05) and ** is a highly significant difference (*p* < 0.01).

**Table 1 vetsci-11-00283-t001:** Primer information of validated and internal reference genes.

Genes Name	GenBank Accession Number	Primers Sequence (5′-3′)	Product Length
*ARRB2*	XM_021067791	F: TAAAGTGGATCCTGTGGATG R: TAGGTGGCGATGAACAGGTC	136 bp
*AR*	NM_214314	F: CAGCCTTCACAACAGCAATC R: CAGCTGAGTCATCCTCGTCC	116 bp
*CYP11A1*	NM_214427	F: TCCACCCCATCTCCGTGAC R: CCCAGCCAAAGCCCAAGTT	213 bp
*CYP17A1*	NM_214428	F: AAGCCAAGACGAACGCAGAAAG R: AGATGGGGCACGATTGAAACC	228 bp
*EGFR*	NM_214007	F: CAAGTAACAAGCTCACCCAG R: ACCTCCTGAATGGTCTTTAG	134 bp
*ETS1*	NM_001162886	F: CGACAGGGACCTCTCAACTC R: AGTGGGACATCTGCACATTC	150 bp
*HSD17B1*	NM_001128472	F: TGATAGGGCTTCCCTTCAAC R: GCTCGAAGTGGCTCAGGTAG	246 bp
*IGF1R*	NM_214172	F: ACATCCTGCTCATCTCCAAG R: AGATGACCAGGGCGTAGTTG	160 bp
*INHBA*	NM_214028	F: CACTCGACGGTCATCAACCA R: GTAAACGTGTCTTCGCGTGG	242 bp
*MIF*	NM_001077213	F: AGAACCGTTCCTACAGCAAG R: CCGTTTATTTCTCTCTTCCG	239 bp
*SCARB1*	NM_213967	F: CGTGGCTCCCAACACCTTAT R: CAAGGGTGCATTGAACCTGC	118 bp
*SMAD4*	NM_214072	F: ATTTCCCTTGCAACATTAGC R: GTGTACAATGCTCAGACAGG	145 bp

**Table 2 vetsci-11-00283-t002:** Statistics of the sequencing samples of Tibetan pigs (TPs) and Yorkshire pigs (YPs).

Samples Name	Total Number of Clips	Mapping Segments	Mapping Fragment SCALE	Single Entry Mapping (math.)	Singles Mapping Ratio	Pairwise Mapping	Pairwise Mapping Ratio
TP1	45,605,842	43,378,626	95.12%	16,678,059	36.57%	25,572,880	56.07%
TP2	42,150,818	39,992,833	94.88%	15,590,339	36.99%	23,155,796	54.94%
TP4	44,508,570	4,250,896	95.51%	15,799,744	35.50%	25,263,538	56.76%
TP5	46,849,326	44,357,885	94.68%	17,333,938	37.00%	2,556,750	54.57%
TP6	45,831,932	43,582,914	95.09%	16,385,456	35.75%	26,037,628	56.81%
TP8	45,015,626	42,943,349	95.40%	16,647,796	36.98%	24,972,968	55.48%
YP1	42,171,230	39,970,012	94.78%	16,003,819	37.95%	2,276,000	53.97%
YP4	46,124,354	43,605,829	94.54%	16,731,424	36.27%	25,060,990	54.33%
YP5	44,573,156	4,253,174	95.42%	16,265,637	36.49%	25,169,637	56.47%
YP6	45,240,124	43,370,974	95.87%	18,377,767	40.62%	23,290,138	51.48%
YP7	4,889,036	46,757,012	95.64%	18,687,416	38.22%	2,646,169	54.12%
YP8	43,638,364	41,521,449	95.15%	14,322,046	32.82%	25,469,391	58.36%

**Table 3 vetsci-11-00283-t003:** Detailed information of 12 selected genes.

Gene ID	Gene Name	*p*-Value	Padj	Log_2_FoldChange	Up/Down
enssscg00000012371	*AR*	0.041172381	0.22619568	−1.208248377	up
enssscg00000017918	*ARRB2*	0.020351911	0.148940782	0.750341389	up
enssscg00000025273	*CYP11A1*	0.711936619	0.880393286	0.434294513	down
enssscg00000010591	*CYP17A1*	0.070199787	0.294198628	1.721167825	down
enssscg00000022126	*EGFR*	0.011554751	0.110088481	−0.874207013	down
enssscg00000015235	*ETS1*	0.534482672	0.783558724	−0.235233666	down
enssscg00000035147	*HSD17B1*	0.130743898	0.404312872	1.53080905	up
enssscg00000030560	*IGF1R*	0.007808723	0.088106822	−0.978372778	down
enssscg00000035077	*INHBA*	7.30E−06	0.001399384	2.788332922	up
enssscg00000010067	*MIF*	0.017109167	0.134644053	0.802256275	up
ENSSSCG00000009759	*SCARB1*	0.944557538	0.982148156	0.082004337	up
enssscg00000004524	*SMAD4*	0.023921806	0.162510834	−0.612213548	down

## Data Availability

Data are contained within the article.

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
