# Peer review of "Application of RNA-Seq Technology for Screening Reproduction-Related Differentially Expressed Genes in Tibetan and Yorkshire Pig Ovarian Tissue"

_vetsci, 2024, doi:10.3390/vetsci11070283_

Round 1
Reviewer 1 Report
Comments and Suggestions for Authors
The aim of the study by Yikai Yin and Collaborators presented in this manuscript was to understand, by differential gene expression profiling of ovary tissues, the molecular basis of low infertility traits of Tibetan pig (TO) respect to normal control York pig (YP). The experiments (RNA extraction and quality control, library construction, sequencing, and quality checking) have been performed in the standard and correct way. Standard bioinformatic analyses have been properly applied to sequencing data to obtain and functionally classify a panel of differentially expressed genes between TP and YP ovary tissues. The differentially expressed genes most functionally connected to fertility have been fully discussed. A sufficient sample of DE genes found by the deep sequencing were validated with an alternative technique (RT-qPCR).
I think that the results presented in this manuscript are of some interest. However, Authors should be aware (and discuss this point in the Discussion section), that the differentially expressed genes found with their work are those that exhibit a higher degree of expression difference, and therefore it is only a partial information. In fact, the ovary is a complex organ, composed of various cell types with different phenotypes and functions. A completer and more definitive picture would be obtained by comparing expression profiles at the level of single or groups of ovarian cells with a similar phenotype.
To add a further level of information and thus make the present work more interesting for the scientific community interested in this field, I suggest investigating with bioinformatics tools the structure of the promoters of differentially expressed genes with related function to predict whether specific transcription factors may be involved in their differential expression.
I have a few major and minor points.
Figure 1.
There is not an exhaustive caption for this Figure and therefore many issues remain unclear. There are 16 total RNA profiles displayed, 8 of which are form Tibetan and 8 are from York pig ovaries? Please, clearly indicate which is which.
In some total RNA profiles there are pronounced extra peaks other than those normally expected for 28S, 18S and 5S rRNAs (e.g.: first and second form left in the first row or second from left in the second row); what are those and why are only present in some RNA profiles?
16 RNA samples were obtained but than from 12 of them libraries were produced: what happened to the remaining 4 total RNA samples? The were excluded for quality reasons? Perhaps, it would less confounding if the Authors exclude their profiles from Figure 1.
Figure 2. The description of the graph in Figure 2A is not informative. I understand that a variable number (from 2.607 of YP1 to 10,878 of YP4) are uniquely detected in each library and that 14,404 genes are instead detected (albeit with a different number of reads) in all the 12 libraries. The reads from these common genes are than used for calculating the differential expressed genes as in Figure 2B. Is that correct?
Please explain why you found such degree of variability of gene expression among libraries of the same pig group. It depends on tissue sample variability (in the Materials and Methods is simply stated “At 180 days of rearing, we collected their ovarian tissues”) or on individual differences (age, size, diet, etc.)?
Comparing the heat map of 2B to the graph of DE genes in 2D there is some contradiction. In fact, from the heat map it seems that there are more down-regulated than up-regulated genes in TP versus YP (the opposite appears from Figure 2D).
The entire set of expression data should be added as supplementary Excel files and not only for the 20 most down- or up-regulated genes. Data should be deposited in public repositories reporting the assigned accession numbers in the manuscript.
In the text, please describe how many DE genes were found between the two groups of pig ovaries out of the 14,000 commonly expressed.
In the Supplementary Table: please order genes by negative or positive fold change value. Also, explain what Gene ID without a Gene Name are (non-coding RNA genes, unknown transcripts, etc.).
Author Response
1、Figure 1.There is not an exhaustive caption for this Figure and therefore many issues remain unclear. There are 16 total RNA profiles displayed, 8 of which are form Tibetan and 8 are from York pig ovaries? Please, clearly indicate which is which.
Answer: 12 samples QC peak chart, The first 6 are Tibetan pig samples and the last 6 are York pig samples.
2、In some total RNA profiles there are pronounced extra peaks other than those normally expected for 28S, 18S and 5S rRNAs (e.g.: first and second form left in the first row or second from left in the second row); what are those and why are only present in some RNA profiles?
Answer: The presence of additional significant peaks in the total RNA electrophoretic profile may be the result of factors such as RNA degradation, RNA modification or processing, non-rRNA RNA molecules, sample specificity, etc., and some rna samples undergo these reactions.
3、16 RNA samples were obtained but than from 12 of them libraries were produced: what happened to the remaining 4 total RNA samples? The were excluded for quality reasons? Perhaps, it would less confounding if the Authors exclude their profiles from Figure 1.
Answer: Figure 1 has been modified, 12 samples QC peak plot, 16 samples were sent for testing, 8 Tibetan and 8 York pig samples, of which 4 cases were backup samples, 2 Tibetan and 2 York pig samples, 12 cases were tested and analysed and 12 samples of higher quality were selected. The first 6 were Tibetan pig samples and the last 6 were York pig samples.
4、Figure 2. The description of the graph in Figure 2A is not informative. I understand that a variable number (from 2.607 of YP1 to 10,878 of YP4) are uniquely detected in each library and that 14,404 genes are instead detected (albeit with a different number of reads) in all the 12 libraries. The reads from these common genes are than used for calculating the differential expressed genes as in Figure 2B. Is that correct?
Answer: Correct, a total of 14,404 co-expressed differential genes were found in a total of 12 libraries.
5、Please explain why you found such degree of variability of gene expression among libraries of the same pig group. It depends on tissue sample variability (in the Materials and Methods is simply stated “At 180 days of rearing, we collected their ovarian tissues”) or on individual differences (age, size, diet, etc.)?
Answer: Differences in gene expression between different libraries from the same pig population may be influenced by a combination of factors. Tissue sample variability leads to the appearance of differences.
6、Comparing the heat map of 2B to the graph of DE genes in 2D there is some contradiction. In fact, from the heat map it seems that there are more down-regulated than up-regulated genes in TP versus YP (the opposite appears from Figure 2D).
Answer: A cursory look at the heatmap is that there appear to be more down-regulated genes than up-regulated genes, but heatmaps and histograms have different emphases when presenting data. Heatmaps are typically used to show patterns and trends in a large number of data points, whereas bar charts are more specific about the number or changes in a particular category. So it turns out to be right, the up-regulated genes are greater than the down-regulated genes. The entire set of expression data should be added as supplementary Excel files and not only for the 20 most down- or up-regulated genes. Data should be deposited in public repositories reporting the assigned accession numbers in the manuscript.
Answer: Data already stored in public repositories.
7、In the text, please describe how many DE genes were found between the two groups of pig ovaries out of the 14,000 commonly expressed.
Answer: Of the 14,000 commonly expressed genes, 651 differential genes were identified between the two groups of pig ovaries, as described in the text.
8、In the Supplementary Table: please order genes by negative or positive fold change value. Also, explain what Gene ID without a Gene Name are (non-coding RNA genes, unknown transcripts, etc.).
Answer: The Supplementary Table has sorted the genes by negative or positive fold change values, and a gene ID without a gene name refers to a unique identifier assigned to a gene in the genome database is also an identifier for an unknown transcript.

Reviewer 2 Report
Comments and Suggestions for Authors
This study provides a beneficial analysis of the low fertility issue in Tibetan pigs and constructs a transcriptome map of ovarian tissue from Tibetan and Yorkshire pigs using RNA-Seq. By screening and validating candidate genes closely related to reproductive traits, it has potential value for understanding the reproductive characteristics of Tibetan pigs and may provide references for future molecular breeding. However, extensive revisions are still required for the manuscript to facilitate reader comprehension.
1. Unify the use of qRT-PCR and RT-qPCR throughout the text, with the full name required only upon the first mention.
2. Figure 1 needs to be included in the appendix with improved resolution and should be properly referenced at the corresponding location in the results.
3. There is no correspondence between "Figure" and "Fig." used throughout the text; the labeling of different sub-figures in the text needs to be modified according to the author guidelines and published articles.
4. In the results section "3.5. Enrichment analysis of the KEGG pathway for differentially expressed genes," only the enriched results in the KEGG figure or the significantly differential and reproductive trait-related pathway results in the supplementary table should be listed; however, pathways such as the "PI3K-Akt signaling pathway" cannot be found throughout the text; in addition, the q-values in the KEGG enrichment results are all greater than 0.05 and need to be carefully checked; there are some errors in punctuation and repeated words in this paragraph that need to be carefully reviewed throughout the text to ensure correctness.
5. The title of Figure 5 is too simplistic and should be modified with reference to published RNA-seq articles.
6. In the discussion section, examples should be provided to illustrate which pathways or terms enriched by the differential genes are closely related to the reproductive traits of animals.
7. In the conclusion section, a summary is needed of which pathways or terms enriched by the differential genes are closely related to the reproductive traits of animals.
8. In p < 0.05 or 0.01, "p" should be italicized.
Author Response
1.Unify the use of qRT-PCR and RT-qPCR throughout the text, with the full name required only upon the first mention.
Answer: The full text was modified in accordance with the uniform use of RT-qPCR.
2.Figure 1 needs to be included in the appendix with improved resolution and should be properly referenced at the corresponding location in the results.
Answer: Figure 1 in the appendix has been upscaled and the results are cited where appropriate
3.There is no correspondence between "Figure" and "Fig." used throughout the text; the labeling of different sub-figures in the text needs to be modified according to the author guidelines and published articles.
Answer: Subfigures in the text have been modified according to the authors' guidelines.
4.In the results section "3.5. Enrichment analysis of the KEGG pathway for differentially expressed genes," only the enriched results in the KEGG figure or the significantly differential and reproductive trait-related pathway results in the supplementary table should be listed; however, pathways such as the "PI3K-Akt signaling pathway" cannot be found throughout the text; in addition, the q-values in the KEGG enrichment results are all greater than 0.05 and need to be carefully checked; there are some errors in punctuation and repeated words in this paragraph that need to be carefully reviewed throughout the text to ensure correctness.
Answer: Changes have been made to this paragraph. The q-values in the KEGG enrichment results were less than 0.05 when carefully reviewed throughout.
5.The title of Figure 5 is too simplistic and should be modified with reference to published RNA-seq articles.
Answer: Changes have been made.
6.In the discussion section, examples should be provided to illustrate which pathways or terms enriched by the differential genes are closely related to the reproductive traits of animals.
Answer: Changes were made in the discussion section.
7.In the conclusion section, a summary is needed of which pathways or terms enriched by the differential genes are closely related to the reproductive traits of animals.
Answer: Complementary pathways for enrichment of differential genes closely linked to reproductive traits in animals
8.In p < 0.05 or 0.01, "p" should be italicized.
Answer: In p<0.05 or 0.01, ‘p’ has been changed to italics in the text.
Please see the attachment.

Round 2
Reviewer 1 Report
Comments and Suggestions for Authors
The Authors responded to all the critical points I had raised. However, they did not take the suggestion to strengthen their work overall by adding a bioinformatic analysis of the promoters of the genes differentially expressed between the two types of ovaries analysed to highlight common sites of transcription factors potentially involved in differential regulation and therefore in the ovarian phenotype. However, I think that the manuscript is publishable in the presente form.
Author Response
Thank you for agreeing to the publication of this article, and put forward your valuable comments, for your comments I have made the following changes in the text.
Delete seven references according to the content of the article on lines 81, 287, 293, 341, 403, 407, 412 of the article.
The annealing temperature of 56°C is confirmed and labelled in line 160 of the article.
Delete a duplicate sentence in line 85 of the article.
A semantically repetitive sentence was deleted from line 296 of the article.
Line 333 of the article was amended to remove a sentence from the article.
The paragraph in lines 367 to 394 was duplicated and deleted.
The statement on line 422 of the article was deleted.
Modified files have been uploaded within 3 days.
Changes have been made to the version sent by the editor.
References that were not relevant to the content of the manuscript were removed.
Any changes to that manuscript were highlighted.

Reviewer 2 Report
Comments and Suggestions for Authors
1.Confirm again whether the annealing temperature of the primer is 56 degrees?
2. There are several repeated words at line 422 in the conclusion that need to be removed; additionally, please thoroughly check the entire document for any other textual errors.
Author Response

(The authors gave the same response as above.)
